# Effect of Locally Delivered Minocycline on the Profile of Subgingival Bacterial Genera in Patients with Periodontitis: A Prospective Pilot Study

**DOI:** 10.3390/biom12050719

**Published:** 2022-05-18

**Authors:** Toshiya Morozumi, Yohei Nakayama, Satoshi Shirakawa, Kentaro Imamura, Kaname Nohno, Takatoshi Nagano, Haruna Miyazawa, Takahiro Hokari, Ryo Takuma, Shuntaro Sugihara, Kazuhiro Gomi, Atsushi Saito, Yorimasa Ogata, Motohiro Komaki

**Affiliations:** 1Department of Periodontology, Faculty of Dentistry, Kanagawa Dental University, 82 Inaoka-cho, Yokosuka 238-8580, Japan; takuma@kdu.ac.jp (R.T.); sugihara@kdu.ac.jp (S.S.); m.komaki@kdu.ac.jp (M.K.); 2Departments of Periodontology and Research Institute of Oral Science, Nihon University School of Dentistry at Matsudo, 2-870-1 Sakaecho-nishi, Matsudo 271-8587, Japan; nakayama.youhei@nihon-u.ac.jp (Y.N.); ogata.yorimasa@nihon-u.ac.jp (Y.O.); 3Department of Dental Hygiene, Tsurumi Junior College, 2-1-3 Tsurumi, Tsurumi-ku, Yokohama 230-8501, Japan; shirakawa-satoshi@tsurumi-u.ac.jp; 4Department of Periodontology, Tokyo Dental College, 2-9-18 Kanda-Misakicho, Chiyoda-ku, Tokyo 101-0061, Japan; imamurakentarou@tdc.ac.jp (K.I.); atsaito@tdc.ac.jp (A.S.); 5Division of Oral Science for Health Promotion, Faculty of Dentistry & Graduate School of Medical and Dental Sciences, Niigata University, 2-5274 Gakkocho-dori, Chuo-ku, Niigata 951-8514, Japan; no2@dent.niigata-u.ac.jp; 6Department of Periodontology, School of Dental Medicine, Tsurumi University, 2-1-3 Tsurumi, Tsurumi-ku, Yokohama 230-8501, Japan; nagano-takatoshi@tsurumi-u.ac.jp (T.N.); gomi-k@tsurumi-u.ac.jp (K.G.); 7Clinical and Translational Research Center, Niigata University Medical and Dental Hospital, 2-5274 Gakkocho-dori, Chuo-ku, Niigata 951-8514, Japan; haruna-m@dent.niigata-u.ac.jp; 8Division of Periodontology, Department of Oral Biological Science, Niigata University Graduate School of Medical and Dental Sciences, 2-5274 Gakkocho-dori, Chuo-ku, Niigata 951-8514, Japan; hokari@dent.niigata-u.ac.jp

**Keywords:** periodontitis, adjunct treatment, scaling and root planing, local drug-delivery system, minocycline-HCl, subgingival microflora, *Veillonella*, bacterial genus

## Abstract

This prospective pilot study aimed to evaluate the effect of minocycline-HCl ointment (MO), locally delivered as an adjunct to scaling and root planing (SRP), on subgingival microflora. A total of 59 periodontitis patients received SRP as an initial periodontal therapy. In the selected periodontal pockets with probing depths (PD) of 6–9 mm, the sites that exhibited a positive reaction following a bacterial test using an immunochromatographic device were subsequently treated with MO (SRP + MO group, *n* = 25). No additional treatment was performed at sites showing a negative reaction (SRP group, *n* = 34). In addition to subgingival plaque sampling, measurement of clinical parameters including PD, clinical attachment level (CAL), bleeding on probing (BOP), plaque index and gingival index (GI) were performed at baseline and 4 weeks after the initial periodontal therapy. The subgingival microflora were assessed by terminal restriction fragment-length polymorphism analysis. Relative to baseline values, the mean scores for PD-, CAL-, BOP-, and GI-sampled sites were significantly decreased post treatment in both groups (*p* < 0.01). The intra-comparisons showed a significant decrease in the counts of the genera *Eubacterium*, *Parvimonas*, *Filifactor*, *Veillonella, Fusobacterium*, *Porphyromonas*, *Prevotella*, and unknown species in the SRP + MO group (*p* < 0.05). Inter-comparisons indicated a significant decrease in the genera *Veillonella* in the SRP + MO group (*p* = 0.01). Combination therapy of SRP and local MO induced a change in the subgingival microbial community: particularly, the number of *Veillonella* spp. was markedly reduced.

## 1. Introduction

Periodontitis is an inflammatory disease initiated by pathogenic bacterial biofilms [1]. The pathogenesis of this disease is a consequence of complex interactions between the biofilm and host immunological response that results in dysbiosis of the microbiome and dysregulation of the host inflammatory response [1,2]. It is known that approximately 500 distinct microbial species are present in subgingival plaque [3]. However, studies using 16S ribosomal DNA (rDNA) amplification have revealed that most subgingival microflora remain uncharacterized [4].

The main objective of periodontal therapy is to promote a shift from a predominantly pathogenic microbial flora to a host-compatible one to achieve clinical and microbiologic stability, resulting in the resolution of inflammation [5]. Scaling and root planing (SRP) is an essential procedure that is frequently performed in the initial phase of periodontal treatment to remove subgingival calculus and biofilm. This treatment has shown excellent clinical results for most patients [6]. However, SRP has some limitations, such as the inability to access deep pockets, root concavities, and furcation areas [7]. Thus, at those sites, it is desirable to introduce adjunctive antimicrobial therapy to reduce or eliminate pathogenic bacteria [8,9].

Several clinical and bacteriological studies have indicated the possibility of various anti-infectious therapies as adjunctive treatments for periodontitis, including disinfectants, photoactivation, laser, and ozone therapy as local application, as well as antimicrobial agents as systemic and local administration [8,9,10,11]. Studies have exhibited the efficacy of systemic administration of antimicrobials in treating periodontal disease [9,12]. However, side effects due to overdose or emergence of drug-resistant bacteria have also been reported [9,13]. Meanwhile, local application of antimicrobial agents has advantages such as independence from patient compliance and maintaining adequate drug concentrations [8,9]. Minocycline is a tetracycline derivative and is active against a wide spectrum of bacteria. It is known that applying minocycline-HCl ointment (MO) as a local drug-delivery system (LDDS) is characterized by features such as marked substantivity, slow-release, superior lipophilic properties, and a therapeutic effect by inhibiting collagenase activity [14,15]. Moreover, minocycline can enhance fibroblast attachment and spreading, which are important for tissue regeneration following periodontal surgery [14]. Thus, MO is one of the more suitable antimicrobial agents for periodontal disease control, especially for local therapy. We previously reported the antimicrobial effects of MO both as a mono and adjunctive therapy [16,17].

Numerous studies have demonstrated the clinical, microbiological, and immunological effects of local antimicrobial administration in the treatment of periodontitis [18]. Most of these bacteriological studies have focused on the number of several specific bacterial species, typified by the red complex [19,20], and knowledge of the contribution of the treatment to the subgingival microbial flora is limited [18,21]. Furthermore, little information is available about the changes in the amount or ratio of bacterial genera in subgingival microbiota in response to combination therapy of SRP and MO [17]. Therefore, it is worth investigating the impact of this combined treatment strategy on the subgingival microbial community.

Taking all these into consideration, we hypothesized that the local application of MO might contribute to reducing the number of obligate anaerobic genera in subgingival microflora. Therefore, the purpose of this study was to investigate the effect of locally delivered MO application as an adjunct to SRP on levels of bacterial genera in the subgingival microflora of patients with periodontitis.

## 2. Materials and Methods

### 2.1. Study Population

This study was a prospective pilot study with a 4-week follow-up period. A total of 59 patients [39 females and 20 males, age range 40 to 80 years; mean age ± standard deviation (SD), 60.9 ± 12.0 years] with generalized moderate-to-severe chronic periodontitis (generalized periodontitis, stage III or IV, grade B) were randomly screened from three facilities in Japan between October 2013 and December 2014 [22,23]. Inclusion criteria consisted of (1) ≥30 years of age, (2) systemically healthy, (3) non-smokers, and (4) possessing at least 20 teeth. Those with the following conditions were excluded: pregnant or breastfeeding, allergy to tetracycline, and use of systemic antimicrobials or anti-inflammatory drugs within 3 months before enrollment. Individuals who had received periodontal treatment within the previous 6 months were also excluded.

This study was conducted in compliance with the principles of the Helsinki Declaration. Written informed consent was obtained from all participants involved. The protocol was approved by the Ethics Committee of each study center and was registered within a clinical trials database (UMIN000011943). Participants were assigned code numbers to identify them throughout the study period.

### 2.2. Clinical Protocol

Experimental procedures and data collection were carried out from December 2013 to February 2015. The study flowchart is shown in Figure 1. Based on a full-mouth periodontal examination at the first visit, a periodontal pocket at a site maximum per individual (probing depth (PD) 6–9 mm) was selected for sampling and measurements.

Approximately 1 month before the study, all participants received standard oral hygiene instructions using a toothbrush, interdental brush, and dental floss, according to movies for the fundamental practice of periodontology produced by the Japanese Society of Periodontology. Over the course of several visits, they received full-mouth supragingival scaling. All participants eventually achieved effective individual plaque control record of <20%. To determine baseline parameters, collection of subgingival plaque and periodontal examination at the selected pockets were carried out. Subsequently, SRP using both hand (Gracey curettes, Hu-Friedy, Chicago, IL, USA) and ultrasonic instrumentation (Solfy F; Morita, Co., Kyoto, Japan) with fine tips under local anesthesia with 2% lidocaine (plus 1:80,000 epinephrine) was performed at all sites with a PD ≥ 4 mm over an average period of 10 weeks (range 8–12 weeks). A bacterial test of subgingival plaque from the selected sites was performed using an immunochromatographic device (DK13-PG-001, detection limit: 10^4^) to detect *Porphyromonas gingivalis* 4 weeks after the final SRP [24,25]. Participants with a site that exhibited a positive reaction (*n* = 25) by bacterial test were assigned to the SRP + MO group (19 females and 6 males, age range 41 to 78 years; mean age ± SD, 60.6 ± 13.6 years), and underwent LDDS treatment using MO four times over an average period of 6 weeks (range 4–8 weeks). Briefly, 2% minocycline-HCl gel (Periofeel; Showa Yakuhin Kako Co., Ltd., Tokyo, Japan) was gently inserted into the base of the periodontal pocket and then slowly pulled out in a zig-zag motion while continuing the injection, as described previously [25,26]. No additional treatment was performed at sites showing a negative reaction (*n* = 34, SRP group: 20 females and 14 males, age range 40 to 80 years; mean age ± SD, 61.2 ± 10.9 years). A second round of subgingival plaque collection and periodontal examination was performed 4 weeks after the final application of MO to reassess initial periodontal therapy.

### 2.3. Clinical Assessment

Five clinical parameters were recorded using a periodontal probe (PCP-11, Hu-Friedy) based on periodontal examination: bleeding on probing (BOP), PD, and clinical attachment level (CAL) at six sites per tooth; plaque index (PlI) and gingival index (GI) at four sites per tooth. The BOP value was calculated as BOP (+) = 1 and BOP (−) = 0. These data were used for diagnosing periodontitis and as the secondary endpoint. Calibration by four examiners who were periodontists was performed using two different types of periodontal disease models (P15FE-500HPRO-S2A1-GSF, P15FE-500HPRO-S2A1-GSD; NISSIN, Kyoto, Japan) before the start of the study. Full-mouth PD and recessions were measured twice, and the intra-examiner repeatability of CAL was assessed. The examiner’s outcomes were judged to be reproducible after reaching a percentage of agreement within ±1 mm between repeated measurements of at least 95%.

### 2.4. Sample Collection

After removal of the supragingival plaque, two sterile number-40 paper points (Zipperer Absorbent Paper Points; VDW GmbH, Munich, Germany) were inserted consecutively into the periodontal pocket for 10 s per point to collect a subgingival plaque sample. The plaque samples were immediately sent to TechnoSuruga Laboratory Co. Ltd. (Shizuoka, Japan) for terminal restriction fragment length polymorphism (T-RFLP) analysis, which is a culture-independent, rapid, sensitive, and reproducible method of assessing diversity of complex bacterial communities [27,28,29].

### 2.5. T-RFLP Analysis

Amplification of 16S rRNA gene sequences of the extracted DNA samples was performed using the following universal primers: 27F (5′-AGAGTTTGATC CTGGCTCAG-3′) and 1492R (5′-GGTTACCTTGTTACGACTT-3′), as previously described [30,31,32]. Primer 27F was labeled at the 5′-end with 6′-carboxyfluorescein (Applied Biosystems Japan, Ltd., Tokyo, Japan). Polymerase chain reaction (PCR) amplicons were digested with 10 U of *Msp*I (Takara Bio Inc., Kusatsu, Japan), and the resultant terminal restriction fragments (T-RFs) were analyzed by capillary electrophoresis using an ABI PRISM 3130*xl* Genetic Analyzer (Applied Biosystems) [33]. GeneMapper^®^ software (Applied Biosystems) was used to estimate the fragment sizes. The T-RFs with peak heights <50 fluorescence units were excluded from the analysis. Fragments were resolved to one base pair by manual alignment with size-standard peaks from different electropherograms. The predicted T-RFLP patterns of the 16S rDNA of known bacterial species were obtained using computer-simulated T-RF data [31,32]. Similarities in microbial patterns among the samples were elucidated using cluster analyses (GeneMaths; Applied Maths, Sint-Martens-Latem, Belgium). The results were arranged to produce a dendrogram. The dendrogram type was established by Pearson’s similarity coefficient analysis and the unweighted pair group method with the arithmetic mean [34]. The number (log_10_) of detected bacteria at the genus level was computed by multiplying each peak area ratio (PAR) by the total bacterial counts, as quantified by PCR analysis. We used these analyses as primary endpoints in the study.

### 2.6. Quantification of Periodontal Bacteria

The remaining bacterial DNA samples were submitted for quantitative analysis of periodontal pathogens at BML, Inc. (Tokyo, Japan). Total bacterial and periodontopathic bacterial counts, including *P. gingivalis*, *Aggregatibacter actinomycetemcomitans*, *Prevotella intermedia*, and *Tannerella forsythia*, were performed using a modified Invader PLUS assay, as described previously [35,36,37,38].

### 2.7. Statistical Analysis

A descriptive analysis was conducted (means ± SD) for the collected data. Intragroup comparisons of clinical parameters and bacterial levels between two time points were performed using the Wilcoxon signed-rank test. All intergroup comparisons were conducted using the Mann—Whitney U test. A comparison of the rates of the increase or decrease in the proportion of each bacterial genus between groups after treatment was verified by the chi-square test. A probability (*p*) value < 0.05 was considered statistically significant. All analyses were conducted using IBM SPSS Statistics V19 software (IBM Japan, Tokyo, Japan).

## 3. Results

All participants successfully completed the study protocol. None of the participants reported any general or oral health problems during the study period. The ratio of tooth types (single/double rooted teeth) selected for the study in the SRP + MO group and SRP group was 10/15 and 14/20, respectively.

### 3.1. Periodontal Parameters

Table 1 shows the results of intra- and inter-group comparisons of the clinical parameters. Relative to baseline values, the mean scores for PD-, CAL-, BOP-, and GI-sampled sites were significantly decreased post treatment in both groups (*p* < 0.01). Meanwhile, the inter-comparison results showed no significant differences in parameters from baseline to post treatment. The mean scores of full mouth-PD and -BOP post treatment were significantly decreased compared with baseline values (full mouth-PD: 3.25 ± 0.73 to 2.79 ± 0.72, full mouth-BOP: 0.23 ± 0.15 to 0.11 ± 0.10; *p* < 0.0001) in the SRP + MO group. Similar results were observed in the SRP group (full mouth-PD: 3.15 ± 0.43 to 2.52 ± 0.42, full mouth-BOP: 0.24 ± 0.17 to 0.10 ± 0.07; *p* < 0.0001). The inter-comparisons also showed no significant differences in both parameters from baseline to post treatment.

### 3.2. Increase and Decrease in Subgingival Bacterial Genera

A comparison of the rates of increases or decreases in the PAR of each bacterial genus after initial periodontal treatment in both groups is presented in Table 2. Except for those that had no change, even a modest change was classified as an increase or decrease. The number indicates the sites with a post-treatment increase or decrease in the PAR of bacterial genera. There were significant differences in changes in the proportions of *Veillonella* and *Neisseria* spp. between groups (*p* < 0.05).

### 3.3. Subgingival Bacterial Genera

Table 3 shows the results of the intra- and intergroup comparisons of subgingival bacterial counts of total bacteria quantified by PCR and several bacterial genera identified by T-RFLP analysis. The number (log_10_) of detected bacteria at the genus level was computed by multiplying each PAR by the total bacterial count. Relative to the baseline values, the numbers of the genera *Eubacterium*, *Parvimonas*, *Eubacterium*, *Filifactor*, *Fusobacterium*, *Porphyromonas*, *Prevotella*, and unknown species were significantly decreased post treatment in both groups (*p* < 0.05). Counts of *Eubacterium* and *Veillonella* spp. were significantly decreased relative to baseline levels post treatment only in the SRP + MO group (*p* < 0.05). The number of *Eubacterium* and *Filifactor* spp. at baseline in the SRP + MO group was greater than that in the SRP group (*p* < 0.01). Post-treatment counts of *Veillonella* spp. were significantly lower in the SRP + MO group than those in the SRP group (*p* = 0.01).

### 3.4. Periodontopathic Species

Intra- and intergroup comparisons of subgingival bacterial counts of primary periodontopathic species at baseline and post initial periodontal treatment are shown in Table 4. Relative to the baseline values, there was a significant reduction in *P. gingivalis* counts after treatment in both groups (*p* < 0.0001). In both groups, *P. intermedia* and *T. forsythia* counts were also significantly reduced relative to the baseline post treatment (*p* < 0.05). Significantly lower counts of *P. gingivalis* and *P. intermedia* at baseline in the SRP group compared with that in the SRP + MO group were observed (*p* < 0.05). Likewise, *P. gingivalis* counts post treatment in the SRP + MO group were significantly lower than those in the SRP group (*p* < 0.01).

## 4. Discussion

In the present study, we examined the bacteriological effects of the local application of MO as an adjunct to SRP on levels of bacterial genera in the subgingival microflora of patients with periodontitis. The results indicated that the combination therapy significantly reduced the counts of most obligate anaerobic bacterial genera, including *Porpyromnas* and *Veillonella* spp., whereas it did not change the counts of facultative anaerobic genera such as *Streptococcus* and *Neisseria* spp. This may be because mechanical removal and local antibiotic administration reduced most of the bacteria, but the change from anaerobic conditions in the subgingival pocket allowed the recovery of the facultative anaerobes and stabilized the subgingival environment in the SRP + MO group. While, in the SRP group, counts of *Veillonella* spp. did not decrease as in the case of other obligate anaerobes, but rather tended to increase. The genus *Veillonella* are anaerobic Gram-negative cocci lacking a flagellum, spore, and capsule. It is known that *Veillonella* spp. plays a pivotal role as an early colonizer in the establishment of multispecies subgingival biofilm communities [39,40]. Furthermore, approximately 10 weeks had elapsed since the final SRP treatment in the SRP group. Hence, it is presumed that counts of *Veillonella* spp. recovered as well as *Streptococcus* spp. due to new plaque formation. In the SRP + MO group, there was a trend toward a greater reduction in the number of *Eubacterium*, *Parvimonas*; *Eubacterium*, *Filifactor*, and *Fusobacterium* spp. than in the SRP group. Further validation of their responsiveness to MO may be needed.

The numbers of the four selected subgingival bacterial species were significantly decreased after treatment in both groups, except for *A*. *actinomycetemcomitans*, consistent with previous reports [17,20]. This outcome at the species level is mostly consistent with that at the genus level, which exhibited a decrease in counts of *Porphyromonas* and *Prevotella* following treatment. In particular, the amount of *P. gingivalis* was markedly reduced in the SRP + MO group. This suggests a high response of *P. gingivalis* to this combination therapy.

Among the antimicrobial agents that can potentially be used, Minocycline-HCl has been chosen for periodontal treatment. It has a wide spectrum of action against aerobic and anaerobic bacteria, including major periodontal pathogens [14,15]. Various reports have described the effect of combination therapy of SRP and MO administration. Goodson et al. reported that the use of MO (a formulation different from the current study) as an adjunct to SRP resulted in significant reductions in the proportions of red-complex bacteria such as *P. gingivalis* and *T. forsythia*. In contrast, the proportions of bacteria such as *Veillonella parvula* and *Streptococcus sanguinis,* increased [41]. Furthermore, Miyazawa et al. recently reported that MO application after subgingival debridement induced an increase in the proportion of *Veillonella* spp. [17]. Regarding the increase in *Streptococcus* and *Veillonella* genera, Teles et al. hypothesized that repeated administration of MO would result in a transient increase in bacterial species resistant to minocycline in subgingival microflora because most species intrinsically resistant to minocycline belong to these genera [42].

As there are no data on bacterial resistance in the study, it is difficult to estimate its impact. There are differences between the aforementioned studies and our study in the condition of the participants. First, the mean PD and CAL of the test sites in the present study were 6.96 and 8.40 mm, respectively. These were approximately 1.5 and 2.5 mm deeper, respectively, than participants in previous studies [17,41]. Second, the treatment phase in the current study was initial periodontal therapy, which is the first application to periodontal pockets, while the participants in Miyazawa et al.’s study were in the supportive periodontal therapy phase and had previously undergone dynamic therapy. Third, the number of periodontopathogenic bacteria was extremely high in the present study. For example, at baseline, *P. gingivalis* and *T. forsythia* were more than 10-fold higher than those in previous studies [17]. Fourth, 40% of the target sites in the present study were double-rooted teeth involving root concavities and/or furcation areas. These suggest that the target sites in this study had advanced periodontitis and the bacterial flora were highly pathogenic. Therefore, we speculate that the change in the counts of facultative anaerobic genera in the SRP + MO group is due to an increase in the number of bacteria that were able to survive under aerobic conditions as a result of the combination therapy, rather than to the effect of resistant bacteria.

Most clinical parameters also drastically improved post treatment in both groups, as in previous reports [18]. However, there was no significant difference in any of the clinical parameters between groups. Several long-term studies have reported that the local application of the MO adjunct with SRP has a sustained effect on clinical parameters compared with SRP alone [20]. In terms of inter-comparison of clinical parameters, the reassessment period may have been too early to detect any significant effect of MO between groups.

This is the first to report changes in the counts of bacterial genera in subgingival microbiota following the combination therapy of SRP and MO implemented in initial periodontal therapy. Trends in bacterial counts and bacterial ratios do not always coincide. For example, if the number for a bacterial genus decreases with treatment, but the total bacterial count decreases substantially more, the ratio of that genus (vs. total bacterial count) will increase. Hence, not only the percentage of each bacterial genus, but also the number of bacteria is important in studying changes in the bacterial flora.

Generally, periodontal surgery has been suggested for residual pockets greater than 5 mm deep after initial treatment [43]. However, considering the systemic influence, it is desirable to avoid periodontal surgery in patients with uncontrolled or severe systemic diseases, such as cardiovascular, blood, hormonal, and neurologic disorders [44]. Therefore, to provide more effective initial periodontal treatment for these patients, it is important to evaluate the efficacy of adjunctive antimicrobial therapy, including MO, even though it should be applied to the limited sites without exceeding the required dosage to prevent the emergence of antimicrobial-resistant bacteria.

The authors are aware of several limitations of this study. The study design is considered as a limitation: a randomized controlled trial using placebo gel or a split-mouth design, might have provided more objectivity. The limited number of subjects analyzed and lack of bias analysis and cytokine profile analysis, are also limitations. In addition, the 4-week follow-up period is relatively very short. A 6-month follow-up might have confirmed a recurrence of the microbial profile. In this study, we used T-RFLP analysis because it is an effective fingerprinting technique for the assessment of the transition of the relative bacterial community, across the large number of samples [45]. However, T-RFLP allows the analysis of the few most abundant members of the community and is not suitable for rigid quantitative analysis. Moreover, a comparison on the efficacy of different antimicrobial agents may provide more pertinent conclusions [46]. In addition, to suppress the chemical-pharmacological action, future studies using probiotics, postbiotics, and paraprobiotics with local action are expected [47,48].

## 5. Conclusions

The present investigation demonstrated that SRP in combination with local MO induced a change in the subgingival microbial community: particularly, the number of *Veillonella* spp. was markedly reduced.

Given that unknown genera accounted for half of the mean PAR in this study, further elucidation of the shift in the subgingival microbiome resulting from antimicrobial therapy by metagenomics analysis using next-generation sequencing should be pursued in future studies.

## Figures and Tables

**Figure 1 biomolecules-12-00719-f001:**
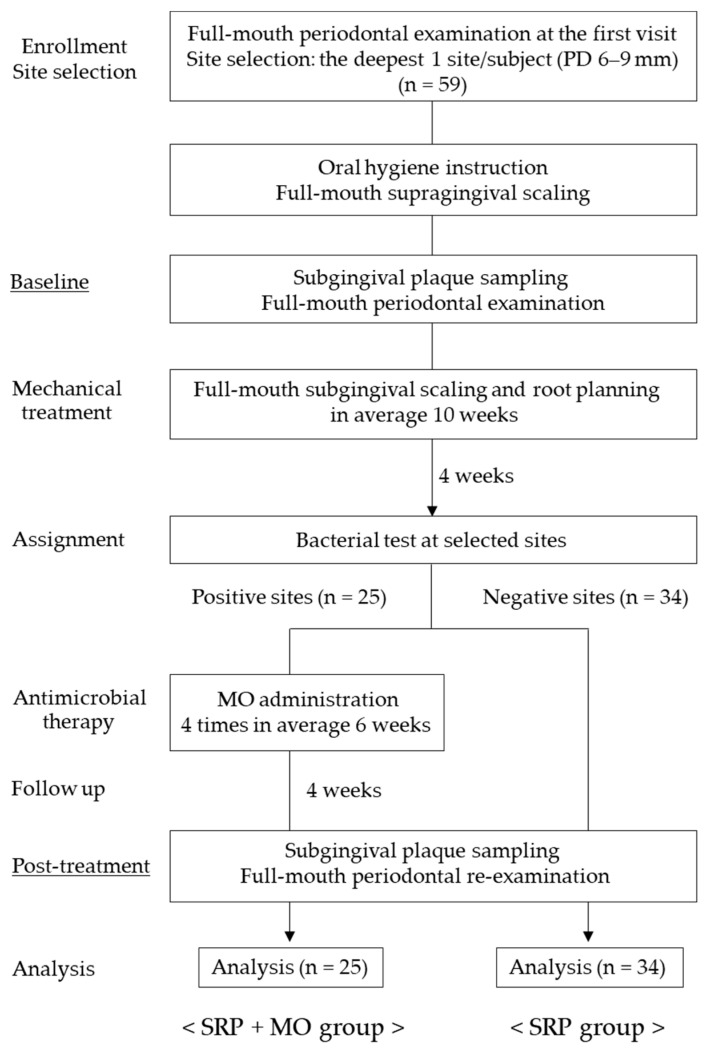
Flowchart of the study from enrollment to completion of the study.

**Table 1 biomolecules-12-00719-t001:** Changes in clinical parameters after initial periodontal therapy.

	SRP + MO Group			SRP Group			Inter-Comparisons	
Mean ± SD	Baseline (*n* = 25)	Post-Treatment (*n* = 25)	Intra-Comparisons (*p*-Value)	Baseline (*n* = 34)	Post-Treatment (*n* = 34)	Intra-Comparisons (*p*-Value)	Baseline (*p*-Value)	Post-Treatment (*p*-Value)
PD-sampled sites (mm)	6.96 ± 0.93	5.04 ± 2.01	<0.0001 *	6.79 ± 0.88	4.15 ± 1.21	<0.0001 *	0.492	0.052
CAL-sampled sites (mm)	8.40 ± 2.25	6.48 ± 2.63	0.001 *	7.88 ± 1.89	5.74 ± 2.18	<0.0001 *	0.377	0.227
BOP-sampled sites *	0.72 ± 0.46	0.36 ± 0.49	0.007 *	0.88 ± 0.33	0.26 ± 0.45	<0.0001 *	0.117	0.436
PlI-sampled sites	0.72 ± 0.68	0.76 ± 0.72	0.763	0.74 ± 0.75	0.62 ± 0.78	0.462	0.987	0.355
GI-sampled sites	1.72 ± 0.46	1.00 ± 0.91	0.002 *	1.76 ± 0.61	0.85 ± 0.86	<0.0001 *	0.314	0.538

Abbreviations: SRP, scaling and root planing; MO, minocycline ointment; PD, probing depth; BOP, bleeding on probing, CAL, clinical attachment level; PlI, plaque index; GI, gingival index. BOP (+) = 1; BOP (−) = 0. The Wilcoxon signed-rank and Mann–Whitney U tests were used for intra- and inter-comparisons, respectively (* *p* < 0.05).

**Table 2 biomolecules-12-00719-t002:** Increases and decreases in the peak area ratio (PAR) of each bacterial genus after initial periodontal treatment (except for those that had no change).

	SRP + MO Group (*n* = 25)		SRP Group (*n* = 34)		
	Increase	Decrease	Increase	Decrease	*p*-Value
*Streptococcus*	19 (76%)	6 (24%)	22 (65%)	12 (35%)	0.352
*Eubacterium*, *Parvimonas*	10 (40%)	15 (60%)	14 (41%)	20 (59%)	0.928
*Eubacterium*, *Filifactor*	4 (16%)	21 (84%)	8 (24%)	25 (74%)	0.478
*Eubacterium*	11 (44%)	12 (48%)	18 (53%)	15 (44%)	0.621
*Veillonella*	10 (40%)	13 (52%)	24 (71%)	10 (29%)	0.033 *
*Fusobacterium*	15 (60%)	10 (40%)	12 (35%)	21 (62%)	0.074
*Porphyromonas, Prevotella*	12 (48%)	13 (52%)	15 (44%)	19 (56%)	0.767
*Neisseria*	18 (72%)	7 (28%)	13 (38%)	19 (56%)	0.018 *
Unknown	11 (44%)	14 (56%)	22 (65%)	12 (35%)	0.113

Some sites exhibited equal PAR values for *Eubacterium*, *Filifactor*, *Eubacterium*, *Veillonella*, *Fusobacterium*, and *Neisseria* spp. Differences between groups were identified by the chi-square test, * *p* < 0.05.

**Table 3 biomolecules-12-00719-t003:** Intra- and intergroup comparisons of subgingival bacterial genera count after initial periodontal treatment.

	SRP + MO Group (*n* = 25)			SRP Group (*n* = 34)			Inter-Comparisons	
Mean ± SD	Baseline	Post-Treatment	Intra-Comparisons (*p*-Value)	Baseline	Post-Treatment	Intra-Comparisons (*p*-Value)	Baseline (*p*-Value)	Post-Treatment (*p*-Value)
Total bacteria (log_10_)	7.45 ± 0.50	6.74 ± 0.51	<0.0001 *	7.24 ± 0.57	6.83 ± 0.58	<0.0001 *	0.163	0.476
*Streptococcus* (log_10_)	5.54 ± 1.78	5.67 ± 0.58	0.253	5.73 ± 1.19	5.55 ± 1.21	0.407	0.753	0.514
*Eubacterium*, *Parvimonas* (log_10_)	5.93 ± 1.36	4.58 ± 2.14	0.001 *	5.91 ± 0.82	5.34 ± 1.18	0.011 *	0.273	0.338
*Eubacterium*, *Filifactor* (log_10_)	6.28 ± 1.42	4.15 ± 2.49	0.0001 *	5.32 ± 2.10	4.12 ± 2.24	<0.001 *	0.006 *	0.600
*Eubacterium* (log_10_)	4.46 ± 2.39	3.68 ± 2.39	0.042 *	4.65 ± 2.05	4.71 ± 1.56	0.367	0.706	0.159
*Veillonella* (log_10_)	4.85 ± 1.95	4.01 ± 2.15	0.019 *	4.31 ± 2.51	5.23 ± 1.19	0.11	0.741	0.01 *
*Fusobacterium* (log_10_)	6.18 ± 0.93	4.92 ± 1.96	0.006 *	5.65 ± 1.62	5.19 ± 1.82	0.031 *	0.223	0.457
*Porphyromonas*, *Prevotella* (log_10_)	6.52 ± 0.71	5.77 ± 0.61	0.001 *	6.38 ± 0.62	5.87 ± 0.64	<0.0001 *	0.361	0.565
*Neisseria* (log_10_)	4.23 ± 2.50	4.76 ± 1.60	0.914	4.78 ± 2.32	4.24 ± 2.27	0.054	0.118	0.951
Unknown (log_10_)	7.07 ± 0.48	6.32 ± 0.55	<0.0001 *	6.84 ± 0.63	6.45 ± 0.59	*p* = 0.001 *	0.145	0.425

Abbreviations: SRP, scaling and root planing; MO, minocycline ointment. Total bacterial counts were calculated by the PCR-invader method. The ratio of each bacterial genus was calculated by T-RFLP analysis. The Wilcoxon signed-rank and Mann–Whitney U tests were used for intra- and inter-comparisons, respectively (* *p* < 0.05).

**Table 4 biomolecules-12-00719-t004:** Intra- and intergroup comparisons of subgingival bacterial counts of primary periodontopathic species after initial periodontal treatment.

	SRP + MO Group (*n* = 25)			SRP Group (*n* = 34)			Inter-Comparisons	
Mean ± SD	Baseline	Post-Treatment	Intra-Comparisons (*p*-Value)	Baseline	Post-Treatment	Intra-Comparisons (*p*-Value)	Baseline (*p*-Value)	Post-Treatment (*p*-Value)
*P. gingivalis* (log_10_)	5.41 ± 1.29	1.64 ± 2.30	<0.0001 *	3.40 ± 2.65	0.48 ± 1.34	<0.0001 *	0.001 *	0.022 *
*A. actinomy*- *cetemcomitans* (log_10_)	0.33 ± 1.17	0.18 ± 0.88	0.593	0.49 ± 1.37	0.38 ± 1.24	0.686	0.64	0.481
*P. intermedia* (log_10_)	3.45 ± 2.70	1.30 ± 2.14	0.001 *	2.39 ± 2.63	1.33 ± 2.13	0.016 *	0.097	1.0
*T. forsythia* (log_10_)	5.59 ± 1.31	3.98 ± 1.65	0.001 *	5.26 ± 1.21	3.31 ± 2.24	<0.0001 *	0.107	0.644

Abbreviations: SRP, scaling and root planing; MO, minocycline ointment. The Wilcoxon signed-rank and Mann–Whitney U tests were used for intra- and inter-comparisons, respectively (* *p* < 0.05).

## Data Availability

All data obtained in this research are described in the manuscript.

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
