# Peer review of "Effect of Locally Delivered Minocycline on the Profile of Subgingival Bacterial Genera in Patients with Periodontitis: A Prospective Pilot Study"

_biomolecules, 2022, doi:10.3390/biom12050719_

Round 1
Reviewer 1 Report
General Comment:
The authors attempted to reveal the effect of minocycline-HCl ointment for periodontal treatment with or without SRP. According to table 1, no significant difference was found between SRP+MO group and SRP only group, but the data suggested some differences in the profile of the bacterial genus. The experimental design of the paper is acceptable. But the authors need to improve the data interpretation and presentation to recommend to the publication.
Specific comment:
T-RFLP:
1) Please add more references to explain general information for T-RFLP analysis.
2) I think the primer pair 27F-1492R is a universal 16S RNA primer to amplify any bacterial 16S RNA. If so, please add a sentence to explain it. If not, please explain what the target gene to amplify is.
Table 2:
I understand the numeric data of the table shows "the number of patients who increases/decrease each genus after treatment. I think If you can convert the data to "percentage," understanding the trend of increases/ decreases of each strain, be much easier.
Table 3:
Can you generate "a heatmap" type figure from the data? If so, you can visualize each genus's number's migration, which is suitable for the reader.
Interpretation of the data:
In the table2, patients increased fusobacterium in the SRP+MO group more than in the SRP group. Do you have any idea why?
Eubacterium and Veillonella are increased in the SRP group but not in the SRP+MO group. Is it caused by MO?
Streptococci increased in both groups. It is thought that SRP changed an anaerobic condition in the subgingival pockets.
The number of Fusobacterium changed before - after treatment in the SRP-MO group is more drastically than in the SRP-only group. Eubacterium and Eubacterium paravimonas too.
Other Misc comment:
Please discuss the Pros/Cons of using T-RFLP to use this kind of research. Also, add the information about the spectrum of minocycline to the discussion.
Author Response
Authors’ responses to the reviews’ comments
We thank the editor and the reviewers for your valuable comments. We provide point-by-point responses as follows. We revised our manuscript accordingly (sky-blue: reviewer 1, pink: reviewer 2, yellow: reviewer 3, green: reviewer 4).
Reviewer 1
T-RFLP:
Question 1) Please add more references to explain general information for T-RFLP analysis.
Responses: Thank you for your advice. Four references with general information for T-RFLP were added in the Materials and Methods section.
Question 2) I think the primer pair 27F-1492R is a universal 16S RNA primer to amplify any bacterial 16S RNA. If so, please add a sentence to explain it. If not, please explain what the target gene to amplify is.
Responses: Thank you for your indication. We added a sentence to explain it.
Table 2:
Question 3) I understand the numeric data of the table shows "the number of patients who increases/decrease each genus after treatment. I think If you can convert the data to "percentage," understanding the trend of increases/ decreases of each strain, be much easier.
Responses: Thank you for your kind advice. The converted % is also noted side by side.
Table 3:
Question 4) Can you generate "a heatmap" type figure from the data? If so, you can visualize each genus's number's migration, which is suitable for the reader.
Responses: We thank the reviewer for the valuable suggestion. At this time, it is difficult to generate a heatmap figure for the data. We feel that the table format is best suited to present our data. However, we will take this into account in our future studies.
Interpretation of the data:
Question 5) In the table2, patients increased fusobacterium in the SRP+MO group more than in the SRP group. Do you have any idea why?
Responses: We speculate that this is because there are more participants in the SRP + MO group who have a smaller reduction rate of Fusobacterium spp. than that of the total bacterial counts, resulting in a higher composition ratio of Fusobacterium spp.
Question 6) Streptococci increased in both groups. It is thought that SRP changed an anaerobic condition in the subgingival pockets.
Responses: Thank you for your advice. We stated it in the Discussion.
Question 7) Eubacterium and Veillonella are increased in the SRP group but not in the SRP+MO group. Is it caused by MO?
Responses: Thank you for your accurate indication. The decrease in bacterial counts in the SRP+MO group may be due to the effect of MO administration. On the other hand, in the SRP group, Eubacterium and Veillonella did not decrease as in the case of other obligate anaerobe, probably because the composition of the bacterial flora changed under the influence of SRP, as in the case of Streptococci and other facultative anaerobe. We stated it in the Discussion.
Question 8)) The number of Fusobacterium changed before - after treatment in the SRP-MO group is more drastically than in the SRP-only group. Eubacterium and Eubacterium paravimonas too.
Responses: Thank you for your accurate advice. We stated it in the Discussion.
Other Misc comment:
Question 9) Please discuss the Pros/Cons of using T-RFLP to use this kind of research. Also, add the information about the spectrum of minocycline to the discussion.
Responses: We appreciate these important suggestions. We described Pros/Cons of T-RFLP in the limitation description in the Discussion section. As for the spectrum of minocycline-HCl, we added such information in the Discussion section.

Reviewer 2 Report
Dear Authors,
After a careful analysis of “Effect of Locally Delivered Minocycline on the Profile of Subgingival Bacterial Genera in Patients with Periodontitis: A Prospective Pilot Study” manuscript, I want to congratulate you for your hard work. I consider the manuscript suitable for publication, after addressing the following aspects:
- Please, detail the recommended oral hygiene measures. Did they imply antiseptic oral rinses?
- Lines 120-123 need to be rephrased.
- Please, check the word order in sentences throughout the manuscript.
Author Response
Authors’ responses to the reviews’ comments
We thank the editor and the reviewers for your valuable comments. We provide point-by-point responses as follows. We revised our manuscript accordingly (sky-blue: reviewer 1, pink: reviewer 2, yellow: reviewer 3, green: reviewer 4).
Reviewer 2
Question 1): Please, detail the recommended oral hygiene measures. Did they imply antiseptic oral rinses?
Responses: Thank you for your indication. We have included specific oral hygiene instructional content in the manuscript. Due to the impact on the data, antiseptic oral rinses were not used during the course of this study.
Question 2): Lines 120-123 need to be rephrased.
Responses: Thank you for your indication. We rephrased the sentence for clarity.
Question 3): Please, check the word order in sentences throughout the manuscript.
Responses: Thank you for your indication. We confirmed the word order in sentences throughout the manuscript.

Reviewer 3 Report
The authors propose a pilot study that analyses the effect of a locally applied minocycline-HCl ointment, as an adjunct to scaling and root planing on subgingival microflora. The authors measured both clinical parameters and the subgingival microflora at baseline and one month after therapy. The authors concluded that combination therapy with SRP and local MO induced a change in the subgingival microbial community.
The subject is not new, and the authors do not offer a long follow-up period.
Besides this, there are several other issues that should be addressed. Please see the enclosed PDF for further details.

Author Response
Authors’ responses to the reviews’ comments
We thank the editor and the reviewers for your valuable comments. We provide point-by-point responses as follows. We revised our manuscript accordingly (sky-blue: reviewer 1, pink: reviewer 2, yellow: reviewer 3, green: reviewer 4).
Reviewer 3
Abstract:
Question 1) The authors should specify in the abstract which clinical parameters were investigated.
Question 2) The exact time should be specified – after 4 weeks
Question 3) Too vague, please give more details.
Question 4) Which anaerobic bacteria?
Question 5) The conclusion is too vague. Please reformulate.
Responses: Thank you for your indication. We have followed your suggestions and corrected the descriptions to be clear.
Introduction:
Question 6) The authors should specify other locally administered therapies, such as photoactivation, laser, ozone therapy. I suggest: Martu, M.-A et al. Antioxidants 2021 and Nicolae, V et al. Rev. Chim.(Bucharest), 2015.
Question 7) The authors should also specify that minocycline may aid in faster periodontal tissue regeneration.
Responses: Thank you for your advice. We added them in the Introduction.
Question 8) What is the originality of this study? What question in the literature are you trying to answer?
Responses: Thank you for your comment. We re-emphasized the originality of this study in the Discussion section. As for the question we were trying to answer, it is shown in the Introduction (please see the lines highlighted in yellow in the revised manuscript).
Materials and Methods:
Question 9) Was power analysis performed to determine the minimum group size?
Responses: Sample size calculation was not performed because group assignment was based on the results of bacteriological tests after SRP. Also, this trial was a pilot study.
Question 10) Was bias analysis performed?
Responses: We did not perform bias analysis. It was described as one of limitations.
Discussion:
Question 11) The discussions section is severely lacking in comparisons with other studies, this is not a new subject in the literature and the authors should compare their results more thoroughly to those of other researchers.
Question 12) The authors must specify the originality of the study. It is not clear what new information this study brings.
Responses: Thank you for the advice. We indicated it in the revised manuscript.
Question 13) Unnecessary sentence... what is the relevance in the context of the article?
Responses: Thank you for your indication. We deleted the sentence (The genus Streptococcus, composed of…).
Question 14) More intergroup comparisons should be made. It is not clear whether the authors obtained better results (statistically significant) clinically and microbiologically in the MO + SRP group vs the SRP group.
Responses: Thank you for your advice. We added considerations in various between-group comparisons.
Question 15) The authors should specify that mynocycline is not the only antibiotic that can be used, several other options exist. The authors should specify that in the future a comparison on the efficacy of these different antibiotics should be made so that more pertinent conclusions can be drawn. I suggest: Luchian I, et al. Antibiotics. 2021 Jul;10(7):814.
Responses: We appreciate the advice. Among the antimicrobial agents that can potentially be used, minocycline-HCl has been chosen for periodontal treatment. It has a wide spectrum of action against aerobic as well as anaerobic bacteria, including major periodontal pathogens. Considering advice from you and the reviewer 1, we added such description in the revised manuscript. We also mentioned about the comparison on the efficacy of different antimicrobial agents in future studies, in the limitation statement.
Question 16) This is not the only limitation, the authors should also specify the limited number of subjects analyzed, very short followup period, lack of a split-mouth design, lack of cytokine profile analysis, etc
Responses: Thank you for your indication. We added the issues you pointed out as limitations.
Conclusion:
Question 17) No such conclusions can be made...the authors MUST specify that the follow-up period was only one month, and the literature specifically states that there is a tendency of microbial profile recurrence after 6 months. The minimum follow-up period should have been 6 months.
Responses: Thank you for your indication. We removed the sentence you pointed out and stated regarding short follow-up period as a limitation.

Reviewer 4 Report
Manuscript of considerable interest even if the experimentation is not recent, it needs some revisions.
The keywords are few, to add more specific ones.
Introduction: even if the study was conducted between 2013 and 2014, it would be appropriate to compare the periodontal parameters with the new classification in 2017.
Materials and methods: with which instrumentation was SRP performed?
What home aids did the patients use?
Results: very confusing tables, to make the statistically significant results more evident.
Discussion: to reduce the chemical-pharmacological action, hope for future studies by reducing inflammatory indices and bacterial load with the use of postbiotics, subgingival, probiotics and paraprobiotics with topical action as now studied by the research group of Prof. Scribante et al.
Conclusion: proactive action to be added
Bibliography: references requested in discussions to be added.
Author Response
Authors’ responses to the reviews’ comments
We thank the editor and the reviewers for your valuable comments. We provide point-by-point responses as follows. We revised our manuscript accordingly (sky-blue: reviewer 1, pink: reviewer 2, yellow: reviewer 3, green: reviewer 4).
Reviewer 4
Question 1): keywords are few, to add more specific ones.
Responses: Thank you for your indication. We added more specialized keywords.
Introduction:
Question 2): Even if the study was conducted between 2013 and 2014, it would be appropriate to compare the periodontal parameters with the new classification in 2017.
Response: Thank you for your suggestion. We described each disease names based on the classification used at the time and the new classification.
Materials and methods:
Question 3): With which instrumentation was SRP performed?
Responses: Thank you for your indication. We described specific instrument names in the revised manuscript.
Question 4): What home aids did the patients use?
Responses: Thank you for your indication. We described the contents of home aids used by subjects in the revised manuscript.
Results:
Question 5): Very confusing tables, to make the statistically significant results more evident.
Responses: We apologize for the confusing tables. We corrected the tables to make them more understandable and easier to read.
Discussion:
Question 6): To reduce the chemical-pharmacological action, hope for future studies by reducing inflammatory indices and bacterial load with the use of postbiotics, subgingival, probiotics and paraprobiotics with topical action as now studied by the research group of Prof. Scribante et al.
Responses: Thank you for your indication. Such description with the references suggested was added in the Discussion section.
Conclusion:
Question 7): Proactive action to be added.
Responses: We have described it in the Conclusion section.

Round 2
Reviewer 3 Report
The manuscript has been improved
Reviewer 4 Report
THE MANUSCRIPT HAS BEEN CORRECTLY REVISED, YOU CAN PROCEED WITH THE PUBLICATION